# Impression formation stimuli: A corpus of behavior statements rated on morality, competence, informativeness, and believability

Amy Mickelberg[1☯], Bradley Walker[1☯], Ullrich K. H. Ecker[1☯], Piers Howe[2‡], Andrew Perfors[2‡], Nicolas Fay[1]*

1 School of Psychological Science, University of Western Australia, Perth, Australia, 2 School of Psychological Sciences, University of Melbourne, Melbourne, Australia

☯ These authors contributed equally to this work.
‡ PH and AP also contributed equally to this work.
* nicolas.fay@gmail.com

**Data Availability Statement:** The data and materials are available on the Open Science Framework: https://osf.io/qnv95/.

## Abstract

To investigate impression formation, researchers tend to rely on statements that describe a person's behavior (e.g., "Alex ridicules people behind their backs"). These statements are presented to participants who then rate their impressions of the person. However, a corpus of behavior statements is costly to generate, and pre-existing corpora may be outdated and might not measure the dimension(s) of interest. The present study makes available a normed corpus of 160 contemporary behavior statements that were rated on 4 dimensions relevant to impression formation: morality, competence, informativeness, and believability. In addition, we show that the different dimensions are non-independent, exhibiting a range of linear and non-linear relationships, which may present a problem for past research. However, researchers interested in impression formation can control for these relationships (e.g., statistically) using the present corpus of behavior statements.

## Introduction

Without direct access to the inner thoughts and feelings of others, we often rely on behavioral information to form impressions of people. Some behaviors may elicit a positive impression (e.g., saving a drowning friend), whereas others may elicit a negative impression (e.g., having an extramarital affair). Each behavior serves as a building block in the impression formation process, and the impressions we form guide our social interactions with friends, colleagues, romantic partners, and casual acquaintances [e.g., 1, 2].

To investigate impression formation under controlled laboratory conditions, researchers often present participants with statements that describe a person's behavior (e.g., "Alex ridicules people behind their backs"), and then participants rate their impressions of that person [e.g., 3–5]. Researchers have primarily focused on dimensions that capture important facets of a person's character: (i) morality [also called communion, see 6], encompassing honesty,

**Funding:** The research was supported by a PhD scholarship funded by the Defence Science and Technology Group of the Department of Defence (A.M.) and a Defence Science and Technology (DST) Group Collaborative Agreement 8382 (N.F., B.W., A.P. and P.H.). The funders had no role in the study design, data collection and analysis or decision to publish. They did provide feedback on draft of the manuscript.

**Competing interests:** The authors have declared that no competing interests exist.

loyalty, and cooperativeness, and (ii) competence [also called agency, see 6], encompassing intelligence, efficiency, and capability [7, 8]. It should be noted that some researchers use the label 'warmth' interchangeably with morality [e.g., 7, 9–11] while others argue that warmth is an overarching factor encompassing morality and sociability [12, 13, see also 6]. We follow Brambilla et al. [14, 15] with morality being core to impression formation. Moral behaviors (e.g., "she kept a friend's secret", "he lied to his parents") indicate whether a person's intentions are good or bad, while competence behaviors (e.g., "they achieved a challenging goal", "she did not get good marks at university") indicate their ability to successfully execute a task [11, 16]. Although both dimensions guide impression formation, moral behaviors are found to be more influential than competence behaviors [13, 14, 17, see also 15]. While morality and competence are the major dimensions investigated to date, other dimensions also play a role.

Another dimension that guides impression formation is informativeness. Behavior statements that are high in informativeness are diagnostic of a person's true character, resulting in greater impression change [18, 19]. Research has shown that the informativeness dimension is related to other dimensions: behavior statements that are morally negative are rated as more informative than morally positive statements [e.g., 20, 21] and morally extreme behavior statements are rated as more informative than morally moderate statements [22–24, see 25 for a review]. It has recently been established that the believability of behavioral information is also important to impression formation; person impressions are updated only when the information is considered to be believable, regardless of how informative or extreme the information is [5]. Thus, believability may moderate the effect of the other dimensions known to guide impression formation [see also 26].

To examine how these dimensions inform person impressions, researchers require a corpus of behavior statements that vary on the relevant dimensions [27–30]. To avoid the cost associated with generating a corpus of statements, it is common to use behavior statements that were generated in prior studies [e.g., 5, 17, 19, 31, 32]. However, doing so can be problematic. First, if the behavior statements were rated by a small sample of judges, they may measure the dimensions of interest imprecisely. Second, behavior statements can become outdated, which can make them difficult for participants to evaluate [e.g., whether "replaced the ribbon on his typewriter" indicates competence; see 28] and may limit their contemporary real-world applicability [e.g., whether someone "had difficulty balancing a checkbook" is unlikely to come up in the present day; see 28]. Third, researchers may be interested in dimensions that were not assessed in past studies—for instance, the statements generated by Chadwick et al. [27] and Fuhrman et al. [28] were not rated on informativeness or believability.

To address these issues, we generated a comprehensive and contemporary list of 160 behavior statements that were rated by a large sample of judges ($N = 400$). The statements were rated on four dimensions: morality, competence, informativeness, and believability. In the present study, the behavior statements were designed to vary across the morality dimension (from extreme positive, e.g., "Person X sold their house to fund a local program for the needy", to extreme negative, e.g., "Person X kicked their pet dog hard in the head because it didn't come when called") and the competence dimension (from extreme positive, e.g., "Person X did all the repair work on their car", to extreme negative, e.g., "Person X failed their driver's license test for the fourth time"). This included statements that were designed to be neutral on both dimensions (e.g., "Person X buys a loaf of bread every day, as they love the smell of freshly baked bread in the morning"). We anticipated that the behavior statements would naturally vary on the informativeness and believability dimensions.

We first present the statements and their ratings across the four dimensions of interest. We then examine the relationships between the four dimensions. Any relationships would highlight potential confounds that should be taken into account by researchers. The corpus

provides a normed set of contemporary behavior statements that enables researchers to test new research questions in impression formation.

## Method

The study was conducted in accordance with the National Statement on Ethical Conduct in Human Research [33]. It was approved by the University of Western Australia's Human Research Ethics Office. Participants viewed an approved information sheet before giving informed consent to take part.

### Participants

A convenience sample of participants were recruited from the United States via the online crowd-sourcing platform Prolific. The sample comprised $N = 400$ participants (female: 205; male: 189; other: 5; prefer not to say: 1) with an age range of 18–73 years ($M = 33.66$, $SD = 11.66$). Each participant received the equivalent of £1.50 (approximately US$2) upon completion of the study.

### Behavior statements

The study used a pool of $N = 160$ behavior statements. These included behaviors generated by the authors ($n = 94$), with the remainder ($n = 66$) adapted from prior studies (14,26,27,33). The behavior statements took the form of "Person X. . .", describing a behavior in which Person X is the agent. The behaviors were designed to vary in morality (positive, negative) and competence (positive, negative), including behaviors that tended toward neutral on both dimensions. Moral behaviors were generated with reference to three of the five psychological foundations of morality: harm/care, fairness/reciprocity, and ingroup/loyalty [34, note however that other conceptualisations of morality also exist, e.g., 35, 36].

To help ensure sufficient variation across each dimension, the authors brainstormed statements from five categories: positive morality (48 statements; e.g., "Person X sold their house to fund a local program for the needy"), negative morality (48 statements; e.g., "Person X set fire to the community hall in the middle of the night"), positive competence (20 statements; e.g., "Person X solved a crossword puzzle in the newspaper"), negative competence (20 statements; e.g., "Person X forgot to turn off the bathwater, flooding the house"), and neutral (24 statements; e.g., "Person X went to a friend's house to play a card game"). Statements in the positive and negative competence categories were designed to be neutral on the morality dimension. Although not intentionally designed to vary on informativeness or believability, it was anticipated that the behavior statements would vary on these dimensions.

Each participant was presented with 40 statements selected randomly subject to the following constraints: 12 from each of the positive and negative morality categories, 5 from each of the positive and negative competence categories, and 6 from the neutral category. Pre-testing indicated that participants could become fatigued if they rated more than 40 statements. Behavior statements were sampled such that each statement was rated by 100 participants.

### Procedure

The study was performed online using an internet-enabled device, and took approximately 15 minutes to complete. After participants provided informed consent, they supplied their age and gender, and read over the instructions. The instructions explained that they would rate 40 behavior statements on various dimensions, with each statement describing a different person (e.g., Person 1, Person 2; this reduced the possibility of statements interacting with each other). The

behavior statements were then presented in a random order. Participants rated each statement on its *morality* ("How morally bad or good is the behavior described in the statement?"), from -4 (*very morally bad*) to 4 (*very morally good*), with 0 indicating *neutral*; its *competence* ("How would you rate the person's competence from the behavior described in the statement?"), from -4 (*very incompetent*) to 4 (*very competent*), with 0 indicating *neutral*; its *informativeness* ("How informative is the statement? How valuable is it when forming an impression of the person?"), from 0 (*not informative*) to 8 (*very informative*); and its *believability* ("How believable is the statement? To what extent could it happen in real life?"), from 0 (*not believable*) to 8 (*very believable*). Ratings were entered using horizontally aligned radio buttons. Participants were then debriefed.

## Results

All analyses were performed and all figures created in R [37]. Data visualizations were created using *ggplot2* [38], the *raincloud plot* package [39], and *corrplot* [40]. The data and R Script are available on the Open Science Framework: https://osf.io/qnv95/.

### Preliminary analysis

To identify uniform responding, we calculated each participant's standard deviation across all measures (morality, competence, informativeness, and believability). No outliers were identified using the interquartile rule with a 2.2 multiplier (i.e., cutoff = SD < Q1–2.2 × IQR) [see 41]. In addition, each measure was approximately normally distributed (|skew| < 2 and |kurtosis| < 9).

### Behavior statement ratings

The distributions of the mean morality, competence, informativeness, and believability ratings for each behavior statement are shown in Fig 1. The ratings ranged across the entire morality dimension (Fig 1A) and most of the competence dimension (Fig 1B). Informativeness ratings varied substantially (Fig 1C). Believability ratings varied considerably (Fig 1D), but most statements were rated as believable (i.e., in the upper part of the scale). Ratings for the full corpus of behavior statements are given in Table 1, and an interactive version of Table 1 is available on the Open Science Framework: https://osf.io/jv7fk.

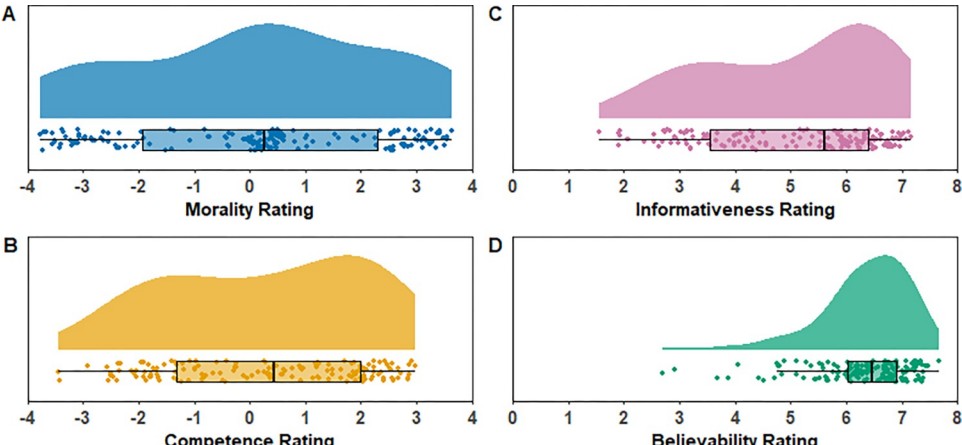

**Fig 1.** Mean morality (A), competence (B), informativeness (C), and believability (D) ratings for each behavior statement. The "cloud" shows the density distribution for the given ratings. Each dot point shows the mean rating for a single behavior statement; points are jittered vertically to avoid overplotting. Boxplots show the first to third quartiles, the bolded vertical line denotes the median, and the whiskers denote 1.5 times the interquartile range.

**Table 1. Mean and standard deviations of morality, competence, informativeness, and believability ratings by behavior statement.**

| Behavior Statement | Morality *M* (SD) | Competence *M* (SD) | Informativen *M* (SD) | Believability *M* (SD) |
|---|---|---|---|---|
| Person X pushed an amputee in front of a train because the amputee made them feel uncomfortable. | -3.78 (0.82) | -2.51 (1.95) | 6.85 (1.78) | 2.69 (2.44) |
| Person X worked in a factory and put broken glass in packets of children's cereal. | -3.72 (1.10) | -2.56 (1.99) | 6.83 (1.79) | 2.92 (2.36) |
| Person X set up a cat trap because they love to catch and torture animals. | -3.70 (0.77) | -1.86 (2.18) | 6.95 (1.60) | 4.77 (2.31) |
| Person X set fire to the community hall in the middle of the night. | -3.67 (0.68) | -1.77 (2.14) | 6.83 (1.65) | 5.11 (2.15) |
| Person X left their children alone in the car for two hours while they went to have a drink on a hot day. | -3.56 (1.31) | -3.45 (1.09) | 7.11 (1.52) | 5.79 (2.08) |
| Person X shook a crying baby so hard while babysitting that it suffered brain damage and nearly died. | -3.55 (1.35) | -3.42 (1.22) | 7.17 (1.30) | 6.02 (2.19) |
| Person X spat in a customer's meal before serving it to him, because the customer had a stutter. | -3.51 (1.31) | -2.54 (1.82) | 6.88 (1.47) | 4.42 (2.53) |
| Person X failed a student because they were African American. | -3.50 (1.28) | -2.92 (1.55) | 7.08 (1.48) | 5.23 (2.20) |
| Person X punched a woman for wearing a hijab because Person X thinks they should not be allowed in this country. | -3.42 (1.30) | -2.38 (1.83) | 6.96 (1.62) | 5.45 (2.17) |
| Person X had an affair with their best friend's wife. | -3.40 (0.89) | -1.68 (1.87) | 6.54 (1.44) | 6.84 (1.52) |
| Person X stole the collection tin of a blind beggar on the street. | -3.38 (1.28) | -1.95 (1.98) | 6.52 (1.83) | 5.37 (2.02) |
| Person X embezzled money from a charity to feed their gambling habit. | -3.36 (1.38) | -2.04 (2.24) | 6.68 (1.64) | 5.84 (1.81) |
| Person X kicked their pet dog hard in the head because it didn't come when called. | -3.31 (1.43) | -2.50 (1.68) | 6.77 (1.57) | 5.82 (2.07) |
| Person X released intimate photos of their ex-partner to their friends and then posted them to the internet. | -3.23 (1.25) | -1.65 (1.92) | 6.72 (1.51) | 6.46 (1.64) |
| Person X could have saved the life of a man stabbed in a dark alley but couldn't be bothered calling an ambulance. | -3.20 (1.29) | -2.35 (1.81) | 6.34 (1.86) | 4.79 (2.31) |
| Person X loosened the wheel nuts of their neighbor's car, because the neighbor always played loud music. | -3.16 (1.45) | -1.10 (2.05) | 6.39 (1.60) | 4.52 (2.24) |
| Person X joined the military because they wanted to see what it was like to kill people. | -3.14 (1.33) | -1.63 (1.77) | 6.48 (1.95) | 4.89 (2.27) |
| Person X regularly cheats on their girlfriend when she is traveling for work. | -3.09 (1.57) | -1.54 (2.11) | 6.79 (1.44) | 7.02 (1.41) |
| Person X started a false rumor that their office colleague Mary used to be a sex worker. | -3.09 (1.25) | -2.12 (1.68) | 6.34 (1.86) | 5.57 (1.96) |
| Person X was in a car accident but drove off before they could see if anyone was injured. | -3.07 (1.62) | -2.45 (1.63) | 6.62 (1.50) | 6.31 (1.72) |
| Person X ignored a new person in the office because they thought the new person was Jewish. | -3.07 (1.34) | -1.91 (1.72) | 6.61 (1.65) | 5.63 (1.94) |
| Person X called a waitress 'dummy' and did not leave a tip, because they didn't like her name. | -2.98 (1.15) | -2.34 (1.46) | 6.72 (1.59) | 5.11 (2.37) |
| Person X started a vicious rumor about their ex-partner, saying that they had neglected their children when they were still married. | -2.97 (1.28) | -1.65 (1.77) | 6.51 (1.53) | 6.57 (1.49) |
| Person X promised to look after their elderly mother's dog but then secretly sold it as soon as their mother moved into the nursing home. | -2.96 (1.35) | -1.31 (1.95) | 6.30 (1.94) | 5.59 (2.16) |
| Person X presented their colleague's idea as their own in order to get a promotion. | -2.86 (1.60) | -1.79 (2.25) | 6.18 (2.07) | 6.28 (1.76) |
| Person X heckled a stranger and made cruel remarks just because the stranger was overweight. | -2.83 (1.64) | -1.88 (1.78) | 6.47 (1.72) | 6.34 (1.78) |
| Person X told a coworker that Person X's brother was HIV-positive, a secret that their brother had told them in confidence. | -2.78 (1.24) | -1.67 (1.51) | 6.06 (1.65) | 6.03 (1.98) |
| Person X said it was their friend who had driven into the neighbor's letterbox even though they did it themselves. | -2.67 (1.22) | -1.73 (1.70) | 5.92 (1.66) | 6.00 (1.71) |
| Person X took performance-enhancing drugs in order to win a race. | -2.64 (1.26) | -1.40 (2.02) | 5.67 (1.90) | 6.68 (1.54) |
| Person X yelled at an elderly person for walking too slowly and being in the way. | -2.64 (1.24) | -1.80 (1.59) | 6.19 (1.56) | 5.85 (1.94) |
| Person X smeared dog poo on their colleague's chair and laughed uncontrollably when their colleague sat on it. | -2.54 (1.50) | -2.05 (1.67) | 6.41 (1.57) | 3.67 (2.26) |
| Person X tears out the last pages of library books to annoy future borrowers. | -2.54 (1.23) | -1.91 (1.70) | 6.25 (1.61) | 4.83 (2.17) |
| Person X scratched their neighbor's expensive car with a key, as he always parked it at the front of Person X's house. | -2.41 (1.44) | -1.00 (1.98) | 5.84 (1.83) | 6.19 (1.76) |

*(Continued)*

**Table 1.** (Continued)

| Behavior Statement | Morality *M* (SD) | Competence *M* (SD) | Informativen *M* (SD) | Believability *M* (SD) |
|---|---|---|---|---|
| Person X left the family business to set up their own business, taking most of the clients with them and causing the family business to go bankrupt. | -2.39 (1.64) | 0.34 (2.38) | 6.24 (1.83) | 5.75 (1.78) |
| Person X found a wallet with $50 in it, took the money out and left the wallet on the floor. | -2.35 (1.88) | -0.67 (1.68) | 6.02 (1.75) | 6.81 (1.54) |
| Person X broke a friend's expensive vase and refused to pay to replace it. | -2.34 (1.29) | -1.71 (1.46) | 5.92 (1.60) | 6.04 (1.68) |
| Person X was on a crowded bus and would not give up their seat to a pregnant woman when asked to. | -2.23 (1.38) | -1.34 (1.52) | 5.94 (1.79) | 6.16 (1.61) |
| Person X closed the elevator door before an elderly neighbor could get in. | -2.08 (1.64) | -0.80 (1.61) | 5.53 (1.81) | 6.06 (1.77) |
| Person X regularly steals office supplies from their job because they feel they deserve it. | -2.01 (1.46) | -0.86 (1.79) | 5.10 (2.01) | 6.68 (1.41) |
| Person X got a friend to fix their gutters and promised guitar lessons in return, but never honored the promise. | -1.96 (1.36) | -1.00 (1.61) | 5.64 (1.65) | 6.12 (1.55) |
| Person X cheated in a card game while playing with a group of their friends. | -1.91 (1.32) | -0.82 (1.73) | 5.16 (1.93) | 6.71 (1.60) |
| Person X pushed in front of another patron in the line at a theater. | -1.80 (1.36) | -1.35 (1.55) | 5.66 (1.76) | 6.09 (2.04) |
| Person X bribed a landlord to be the first to get their apartment repainted. | -1.51 (1.59) | 0.32 (1.72) | 5.29 (1.63) | 5.67 (1.81) |
| Person X pretended to be seriously fouled by an opposing player during a soccer game. | -1.43 (1.44) | -0.54 (1.73) | 5.05 (2.33) | 6.84 (1.38) |
| Person X did not attend their coworker's funeral because they'd had a disagreement before the coworker died. | -1.40 (1.34) | -0.60 (1.43) | 4.92 (1.97) | 5.51 (2.03) |
| Person X burned their country's flag because they don't like their country. | -1.37 (1.70) | -0.92 (1.61) | 5.04 (2.12) | 6.28 (1.90) |
| Person X drove their car the wrong way down a one-way street. | -1.07 (1.35) | -2.39 (1.44) | 4.22 (2.16) | 6.19 (2.01) |
| Person X left their long-term band as soon as an opportunity came up to play with a more successful band. | -0.81 (1.48) | 0.59 (1.43) | 4.88 (2.03) | 6.49 (1.56) |
| Person X forgot it was their wedding anniversary and did not have a gift for their spouse. | -0.76 (1.14) | -1.24 (1.42) | 4.09 (2.07) | 6.73 (1.48) |
| Person X walked into the street without checking for oncoming traffic. | -0.72 (1.15) | -2.32 (1.38) | 4.15 (2.16) | 6.53 (1.57) |
| Person X forgot they had to attend their niece's dance concert that evening. | -0.44 (0.84) | -1.21 (1.21) | 3.43 (1.92) | 6.75 (1.31) |
| Person X forgot to turn off the bath water, flooding the house. | -0.41 (1.26) | -2.09 (1.60) | 3.89 (1.79) | 5.44 (1.93) |
| Person X was forced to pay multiple bank fees for not paying their credit card repayment on time. | -0.40 (0.89) | -1.19 (1.39) | 4.03 (2.01) | 7.08 (1.32) |
| Person X broke off all communication with their family for a while because they had a heated argument with each other. | -0.38 (1.01) | -0.04 (1.08) | 4.44 (1.87) | 6.76 (1.36) |
| Person X forgot to put the alarm on when they were the last one to leave the office. | -0.37 (0.91) | -1.36 (1.18) | 3.54 (1.86) | 6.58 (1.45) |
| Person X forgot to turn the stove off before leaving the house. | -0.31 (0.99) | -1.82 (1.50) | 3.53 (1.83) | 6.37 (1.87) |
| Person X sneezed loudly in an important meeting. | -0.11 (0.51) | -0.12 (0.66) | 1.56 (1.96) | 7.10 (1.34) |
| Person X failed their driver's license test for the fourth time. | -0.07 (0.50) | -2.09 (1.33) | 3.98 (2.17) | 6.27 (1.53) |
| Person X ate their lunch and went back to work with food stuck in their teeth. | -0.02 (0.74) | -0.25 (0.98) | 1.92 (2.01) | 7.28 (1.14) |
| Person X never learned how to ride a bicycle. | -0.01 (0.75) | -0.42 (1.18) | 2.37 (2.20) | 6.36 (1.78) |
| Person X forgot to water their front garden causing the grass to turn brown. | 0.02 (0.65) | -0.99 (1.06) | 2.91 (2.00) | 6.91 (1.63) |
| Person X disappointed their boss when they were unable to attract any new customers. | 0.02 (0.79) | -0.97 (1.33) | 3.52 (2.23) | 6.67 (1.48) |
| Person X went to the supermarket but couldn't remember what they needed to buy. | 0.03 (0.48) | -1.08 (1.27) | 2.93 (2.03) | 6.70 (1.51) |
| Person X went to a fancy restaurant but couldn't pronounce the items on the menu. | 0.06 (0.66) | -0.45 (1.08) | 2.53 (2.15) | 6.75 (1.56) |
| Person X did not meet their sales targets for the month at work. | 0.06 (0.85) | -0.86 (1.09) | 3.12 (2.06) | 7.08 (1.33) |
| Person X arrived at the airport only to discover they had left their passport at home. | 0.06 (0.81) | -1.33 (1.22) | 3.20 (2.04) | 7.03 (1.18) |
| Person X brought in the groceries from the car and dropped one of the bags, which caused the eggs to break. | 0.07 (0.78) | -0.26 (1.12) | 2.06 (2.29) | 7.22 (1.15) |
| Person X can walk on their hands down a flight of stairs. | 0.08 (0.39) | 1.20 (1.66) | 2.62 (2.16) | 4.71 (2.24) |
| Person X ordered a take-away coffee but spilled it when they tried to take a sip. | 0.08 (0.75) | -0.29 (1.19) | 1.90 (2.14) | 7.33 (1.24) |
| Person X solved a crossword puzzle in the newspaper. | 0.11 (0.53) | 1.24 (1.38) | 3.20 (2.24) | 7.26 (1.30) |
| Person X once spent a whole weekend watching furniture restoration videos on the Internet. | 0.15 (0.76) | 0.31 (1.15) | 3.35 (1.98) | 6.00 (2.03) |
| Person X ran out of paint when painting their home and had to go the hardware store. | 0.15 (0.67) | 0.26 (1.40) | 2.54 (2.32) | 7.31 (1.22) |

*(Continued)*

**Table 1.** (Continued)

| Behavior Statement | Morality *M* (SD) | Competence *M* (SD) | Informativen *M* (SD) | Believability *M* (SD) |
|---|---|---|---|---|
| Person X went to the nearby airport, as they like to watch the planes. | 0.15 (0.52) | 0.33 (1.12) | 3.22 (2.22) | 6.04 (1.75) |
| Person X always wins at Trivial Pursuit. | 0.16 (0.68) | 2.08 (1.47) | 3.50 (2.12) | 6.18 (1.89) |
| Person X was the fastest runner when they were in high school. | 0.16 (0.60) | 1.23 (1.42) | 2.79 (2.20) | 6.83 (1.45) |
| Person X locked their keys in the house and had to call a locksmith. | 0.19 (1.04) | -0.48 (1.52) | 3.13 (2.39) | 7.23 (1.20) |
| Person X buys a loaf of bread every day, as they love the smell of freshly baked bread in the morning. | 0.20 (0.97) | 0.28 (1.12) | 2.84 (1.94) | 5.64 (2.27) |
| Person X went to purchase a new pair of shoes but couldn't find any that were comfortable. | 0.21 (0.86) | 0.36 (1.08) | 2.08 (2.40) | 7.07 (1.35) |
| Person X ordered their favorite dish from a Chinese restaurant. | 0.22 (0.86) | 0.47 (1.00) | 1.90 (2.42) | 7.66 (0.99) |
| Person X hailed a bus and asked the bus driver which stop was the closest stop to get to the city. | 0.23 (0.87) | 0.37 (1.61) | 3.09 (2.28) | 6.28 (1.91) |
| Person X has memorized three of Shakespeare's plays. | 0.28 (0.78) | 2.11 (1.43) | 3.98 (2.13) | 5.28 (2.15) |
| Person X likes to go to their local café and sip coffee while reading the newspaper. | 0.29 (0.87) | 0.44 (0.96) | 2.82 (2.42) | 7.43 (1.10) |
| Person X played chess with their friend, winning the game. | 0.30 (0.85) | 1.91 (1.21) | 3.10 (2.21) | 7.37 (1.11) |
| Person X tried to patch a puncture in the wheel of their bike but couldn't, so they purchased a new tube instead. | 0.34 (1.14) | 0.77 (1.80) | 3.25 (2.37) | 7.18 (1.30) |
| Person X prepared a roast chicken and made the stuffing from scratch. | 0.35 (0.99) | 2.01 (1.30) | 3.77 (2.26) | 7.31 (1.17) |
| Person X often sings along to the songs that they are listening to. | 0.36 (1.21) | 0.41 (1.10) | 2.73 (2.47) | 7.25 (1.51) |
| Person X was unable to fix the dripping faucet so they had to call the plumber. | 0.38 (0.87) | 0.03 (1.35) | 2.85 (2.32) | 7.29 (1.27) |
| Person X was running late so they drove to work rather than taking the bus. | 0.40 (1.11) | 0.97 (1.60) | 3.11 (2.42) | 7.07 (1.22) |
| Person X did really well at the quiz night. | 0.41 (1.01) | 2.07 (1.39) | 3.66 (2.05) | 7.27 (1.14) |
| Person X won a weightlifting contest at their gym, after placing second for the last three years. | 0.41 (1.04) | 2.24 (1.36) | 4.32 (2.34) | 6.64 (1.53) |
| Person X won the door prize at the town's community fair. | 0.44 (1.15) | 0.41 (1.09) | 1.92 (2.29) | 6.74 (1.51) |
| Person X went to a friend's house to play a card game. | 0.44 (1.21) | 0.64 (1.30) | 2.70 (2.46) | 7.46 (1.27) |
| Person X won an award for 'Best Newcomer' at a local karaoke event. | 0.45 (1.03) | 1.23 (1.46) | 3.12 (2.31) | 6.70 (1.45) |
| Person X put on a suit and wore their lucky socks in preparation for a job interview. | 0.47 (1.08) | 0.57 (1.34) | 3.43 (2.32) | 6.75 (1.60) |
| Person X learned a secret prize-winning pie recipe from their grandmother before she died. | 0.48 (0.96) | 0.89 (1.17) | 2.97 (2.50) | 6.65 (1.65) |
| Person X was singing loudly to their favorite song in the car. | 0.49 (1.18) | 0.25 (0.93) | 2.91 (2.19) | 7.38 (1.12) |
| Person X accidentally knocked a glass off the table, but managed to catch it before it could smash on the floor. | 0.50 (1.14) | 1.43 (1.44) | 2.72 (2.20) | 6.72 (1.55) |
| Person X learned how to play the piano when they were a child. | 0.51 (1.14) | 1.90 (1.39) | 3.81 (2.29) | 7.38 (1.08) |
| Person X always wrote things down as they would always forget things. | 0.51 (1.08) | 0.93 (1.83) | 4.38 (1.98) | 6.87 (1.45) |
| Person X was able to convince their boss that they were ready for a promotion at work. | 0.52 (0.98) | 2.41 (1.38) | 4.51 (1.90) | 6.57 (1.39) |
| Person X told the children to be quiet in the library. | 0.57 (1.12) | 0.91 (1.22) | 3.49 (2.35) | 7.08 (1.37) |
| Person X is learning French as they always wanted to learn another language. | 0.57 (1.09) | 1.92 (1.28) | 3.99 (2.01) | 7.22 (1.23) |
| Person X did all the repair work on their car. | 0.60 (1.13) | 2.90 (1.16) | 4.56 (1.87) | 6.90 (1.46) |
| Person X ordered pizza while at their friend's farewell party. | 0.61 (1.19) | 0.60 (1.09) | 3.01 (2.20) | 6.41 (1.74) |
| Person X arrived at the art exhibition early so they could view the collection before it got too busy. | 0.62 (1.16) | 1.85 (1.37) | 4.20 (2.20) | 7.13 (1.09) |
| Person X successfully remembered their coworkers' overly complicated coffee orders without writing them down. | 0.79 (1.12) | 2.85 (1.25) | 4.69 (2.04) | 6.38 (1.63) |
| Person X walks to work through the park each day, as they enjoy listening to the birds. | 0.81 (1.30) | 0.91 (1.28) | 4.08 (2.30) | 6.98 (1.32) |
| Person X laughed at a friend's joke even though it wasn't funny. | 0.87 (1.26) | 0.77 (1.24) | 4.31 (2.00) | 6.93 (1.39) |
| Person X cleaned the bookshelf and picked up some items that had dropped onto the floor. | 1.00 (1.21) | 1.23 (1.32) | 3.67 (2.20) | 7.00 (1.39) |
| Person X forgave their partner even though they had been cheating on Person X for two years. | 1.02 (1.95) | -0.24 (1.94) | 5.78 (1.66) | 5.98 (1.88) |
| Person X learned an impressive dance routine in preparation for a friend's wedding. | 1.02 (1.29) | 2.15 (1.37) | 4.32 (2.10) | 6.68 (1.36) |
| Person X received the employee of the month award at their work. | 1.13 (1.48) | 2.67 (1.30) | 5.17 (1.91) | 7.25 (1.23) |

*(Continued)*

**Table 1.** (Continued)

| Behavior Statement | Morality M (SD) | Competence M (SD) | Informativen M (SD) | Believability M (SD) |
|---|---|---|---|---|
| Person X taught their nephew how to drive at the local shopping center car park, after the shops had shut. | 1.43 (1.38) | 1.63 (1.46) | 4.56 (2.27) | 7.01 (1.26) |
| Person X went skydiving despite their intense fear of heights because it was their sister's wish to do it together. | 1.52 (1.34) | 1.18 (1.42) | 5.23 (1.86) | 5.84 (1.72) |
| Person X took their nephew to the fair and bought some cotton candy. | 1.59 (1.28) | 1.13 (1.37) | 4.28 (2.22) | 7.35 (1.16) |
| Person X always pays off their debts first before buying things for themselves. | 1.85 (1.47) | 2.87 (1.50) | 5.72 (1.86) | 6.70 (1.49) |
| Person X regularly volunteers in a town that was exposed to radiation, despite the doctor warning them that their own health would be at risk. | 1.93 (1.51) | 0.15 (2.24) | 5.56 (1.89) | 4.75 (2.12) |
| Person X invited an unpopular coworker to have lunch with them at a new café that had just opened. | 2.06 (1.35) | 1.40 (1.54) | 5.53 (1.85) | 6.15 (1.64) |
| Person X regularly sings at a prison in order to entertain the inmates. | 2.26 (1.32) | 1.71 (1.40) | 5.38 (1.84) | 5.67 (2.05) |
| Person X declined a high-paying job with a weapons manufacturing company because they didn't believe in what the company stood for. | 2.27 (1.76) | 1.49 (1.93) | 5.77 (1.99) | 6.02 (1.79) |
| Person X helped a neighbor move a piano into his second floor apartment. | 2.35 (1.30) | 2.06 (1.43) | 5.34 (1.77) | 6.40 (1.62) |
| Person X shaved their head when they found out their partner had cancer and required radiation therapy. | 2.46 (1.49) | 1.83 (1.63) | 6.29 (1.70) | 7.23 (1.23) |
| Person X put money in the expired parking meter of a stranger. | 2.52 (1.24) | 1.63 (1.57) | 5.90 (1.68) | 6.18 (1.96) |
| Person X circulated a petition in support of civil rights for people in juvenile detention. | 2.54 (1.36) | 2.23 (1.34) | 5.91 (1.63) | 6.46 (1.55) |
| Person X translated the menu items for a foreigner in a restaurant. | 2.54 (1.36) | 2.87 (1.24) | 5.91 (1.73) | 6.78 (1.28) |
| Person X didn't go to a concert they had been looking forward to because their mother was ill. | 2.54 (1.33) | 1.84 (1.44) | 5.80 (1.97) | 6.89 (1.33) |
| Person X took public transport so their sister could use their car to get to work safely. | 2.58 (1.31) | 2.03 (1.46) | 6.03 (1.37) | 6.71 (1.37) |
| Person X called the bank to tell them about money deposited into Person X's bank account by accident. | 2.59 (1.60) | 1.98 (1.70) | 5.85 (1.86) | 5.86 (2.00) |
| Person X stayed back to help a colleague jumpstart their car, even though they then missed the start of a music concert. | 2.61 (1.44) | 2.01 (1.55) | 6.08 (1.79) | 6.70 (1.41) |
| Person X saw someone across the road drop a stack of papers, so they crossed the road to help. | 2.61 (1.14) | 1.80 (1.42) | 5.55 (1.87) | 6.09 (1.70) |
| Person X volunteers at a dog refuge, walking the dogs and cleaning their kennels once a week. | 2.67 (1.14) | 2.06 (1.38) | 5.93 (1.71) | 7.04 (1.21) |
| Person X volunteers to teach English to newly arrived immigrants. | 2.73 (1.25) | 2.72 (1.33) | 6.17 (1.62) | 6.87 (1.28) |
| Person X quit their high-paying job so they could volunteer full time at a nursing home. | 2.78 (1.49) | 1.48 (1.99) | 6.05 (1.73) | 4.05 (2.41) |
| Person X helped paint their neighbor's house even though it was Person X's birthday. | 2.78 (1.21) | 2.20 (1.47) | 6.33 (1.76) | 6.05 (1.90) |
| Person X donates blood once a month even though they have a strong fear of needles. | 2.78 (1.27) | 1.89 (1.59) | 5.99 (1.87) | 6.19 (1.64) |
| Person X put up posters and handed out fliers to help find their neighbor's missing dog. | 2.85 (1.18) | 2.23 (1.38) | 5.97 (1.55) | 7.24 (1.10) |
| Person X offered to let their evicted sister and brother-in-law stay with them for free and sleep in Person X's room while Person X slept on the couch. | 2.86 (1.54) | 1.42 (1.70) | 6.24 (1.59) | 6.25 (1.65) |
| Person X found an expensive briefcase and tried to locate the owner. | 2.92 (1.17) | 1.97 (1.52) | 6.00 (1.50) | 6.23 (1.59) |
| Person X saw a child lost in a supermarket, so they helped find the parents by alerting the staff. | 2.97 (1.30) | 2.57 (1.37) | 6.22 (1.62) | 7.18 (1.24) |
| Person X repaid a loan of $100 that their friend had lent them, even though the friend did not remember it. | 2.98 (1.16) | 2.41 (1.45) | 6.39 (1.60) | 6.91 (1.41) |
| Person X sold their house to fund a local program for the needy. | 3.01 (1.42) | 1.75 (1.86) | 6.15 (1.91) | 3.87 (2.53) |
| Person X drove across the country just to see a friend who had recently lost his wife. | 3.04 (1.10) | 2.00 (1.57) | 6.54 (1.40) | 6.62 (1.44) |
| Person X drove an hour out of their way to pick up a friend and drive him to work because his car had broken down. | 3.08 (1.17) | 2.34 (1.60) | 6.34 (1.63) | 6.79 (1.49) |
| Person X helped their brother renovate his house every night for six months after it had been damaged by fire. | 3.10 (1.20) | 2.68 (1.51) | 6.63 (1.48) | 6.38 (1.59) |
| Person X jumped in to help a friend who was being bitten by a vicious dog, resulting in Person X being seriously mauled. | 3.13 (1.10) | 1.24 (2.07) | 6.42 (1.51) | 6.08 (1.74) |

*(Continued)*

**Table 1.** (Continued)

| Behavior Statement | Morality *M* (SD) | Competence *M* (SD) | Informativen *M* (SD) | Believability *M* (SD) |
|---|---|---|---|---|
| Person X pulled over on a busy highway on a rainy day to help a stranger change his flat tire. | 3.17 (1.00) | 2.57 (1.40) | 6.46 (1.49) | 6.42 (1.75) |
| Person X stepped in when a friend at a pub was getting assaulted for being dark skinned. | 3.18 (1.49) | 2.42 (1.65) | 6.74 (1.39) | 6.81 (1.49) |
| Person X used their body to protect their partner from falling debris during an earthquake. | 3.22 (1.19) | 2.18 (1.67) | 6.48 (1.64) | 6.37 (1.47) |
| Person X hosted a fundraising dinner to raise money for a local homeless shelter. | 3.24 (0.93) | 2.37 (1.36) | 6.47 (1.42) | 6.82 (1.53) |
| Person X worked on a campaign to release wrongfully convicted prisoners. | 3.27 (1.07) | 2.58 (1.48) | 6.45 (1.52) | 6.87 (1.59) |
| Person X risked their life rescuing an animal that was trapped inside a burning house. | 3.28 (1.06) | 2.05 (1.72) | 6.42 (1.51) | 6.33 (1.56) |
| Person X cared for and housed their five nieces and nephews for a year because Person X's sister was very unwell. | 3.36 (1.11) | 2.85 (1.33) | 6.99 (1.15) | 6.65 (1.34) |
| Person X saved a man who was about to be hit by a car by jumping in front of the car and pushing him out of the way. | 3.38 (1.25) | 2.52 (1.81) | 6.75 (1.46) | 5.53 (1.97) |
| Person X offered to pay off the debts of their friend, who had been struggling to pay the bills since their partner died. | 3.41 (1.03) | 2.58 (1.53) | 6.67 (1.47) | 5.66 (2.09) |
| Person X commutes four hours on a bus every week to the local children's hospital, so they can dress as a clown and entertain the children in the cancer ward. | 3.42 (1.03) | 2.31 (1.66) | 6.97 (1.35) | 6.21 (1.72) |
| Person X saw a homeless person in the rain, so they gave the person their jacket and umbrella, plus $20 for a hot meal. | 3.47 (1.03) | 1.99 (1.71) | 6.83 (1.39) | 6.02 (1.90) |
| Person X donated a kidney to a work colleague who would die without it, as they were a perfect match. | 3.51 (1.27) | 2.00 (1.82) | 6.90 (1.47) | 6.16 (1.67) |
| Person X found a wallet containing $1000 and returned it to its rightful owner. | 3.55 (0.86) | 2.54 (1.45) | 6.80 (1.38) | 6.19 (1.70) |
| Person X jumped off a boat to save a drowning friend even though this put Person X's own life at risk. | 3.55 (1.01) | 2.47 (1.58) | 6.95 (1.38) | 6.79 (1.53) |
| Person X turned their home into a shelter for flood victims, making meals and providing clothing to those who needed it. | 3.63 (0.86) | 2.97 (1.23) | 7.03 (1.20) | 6.08 (1.90) |

*Note.* Informativen. = informativeness. Morality ratings varied from -4 (*very morally bad*) to 4 (*very morally good*), competence ratings varied from -4 (*very incompetent*) to 4 (*very competent*), informativeness ratings varied from 0 (*not informative*) to 8 (*very informative*), and believability ratings varied from 0 (*not believable*) to 8 (*very believable*). Means and standard deviations (in parentheses) for each behavior statement are based on 100 ratings. An interactive version of the table is available on the Open Science Framework: https://osf.io/jv7fk.

## Relationships between the dimensions

There were moderate-to-strong Pearson correlations between the morality and competence ratings, the morality and believability ratings, the competence and believability ratings, and the informativeness and believability ratings (see Fig 2).

The relationships between each of the dimensions are visualized in Fig 3. Inspection of the figure indicated linear and non-linear relationships between several pairs of dimensions. We therefore tested for linear and quadratic relationships using orthogonal polynomial regression (see Table 2 for statistical output). The morality and competence ratings showed a strong positive linear relationship, indicating that behavior statements rated as more positive in morality were rated as more competent (see Fig 3A). The morality and informativeness ratings showed a strong quadratic effect, indicating that behavior statements rated as more extreme in morality (negative or positive) were rated as more informative (see Fig 3B). The morality and believability ratings showed both a moderate positive linear relationship and a quadratic relationship (see Fig 3C). The linear effect indicates that behavior statements rated as more positive in morality were rated as more believable, while the quadratic effect indicates that behavior statements rated as more extreme in morality (negative or positive) were rated as less believable.

The competence and informativeness ratings showed a strong quadratic effect (see Fig 3D), indicating that behavior statements rated as more extreme in competence (negative or

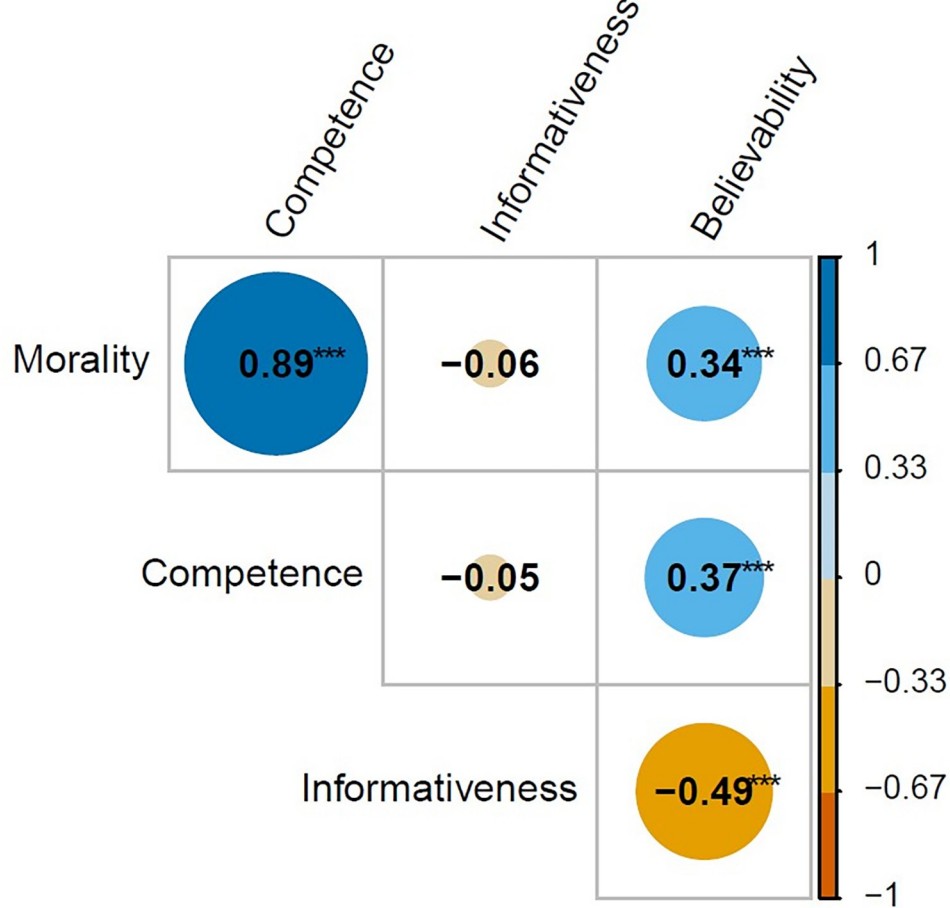

**Fig 2. Correlations between the morality, competence, informativeness, and believability ratings.** The color shows the direction of the relationship, with positive in blue and negative in orange. Circle size shows the strength of the relationship, with a larger circle indicating a stronger relationship. Note $^*p < .050, ^{**}p < .010, ^{***}p < .001$.

positive) were rated as more informative. The competence and believability ratings showed a moderate positive linear relationship and a quadratic relationship (see Fig 3E). The linear effect indicates that behavior statements rated as more positive in competence were rated as more believable, while the quadratic effect indicates that behavior statements rated as more extreme in competence (negative or positive) were rated as less believable. The informativeness and believability ratings showed a strong negative linear relationship, indicating that behavior statements rated as more informative were rated as less believable (see Fig 3F).

## Discussion

The present study provides a normed corpus of 160 contemporary behavior statements. Each behavior statement was rated on the dimensions of morality [11], competence [10], informativeness [23], and believability [5], which are known to affect impression formation. The behavior statement ratings varied widely on the morality, competence, and informativeness dimensions, providing researchers with substantial scope to investigate the effects of these dimensions on impression formation. There was less variation on the believability dimension, with most behavior statements rated as being at least moderately believable. Given that

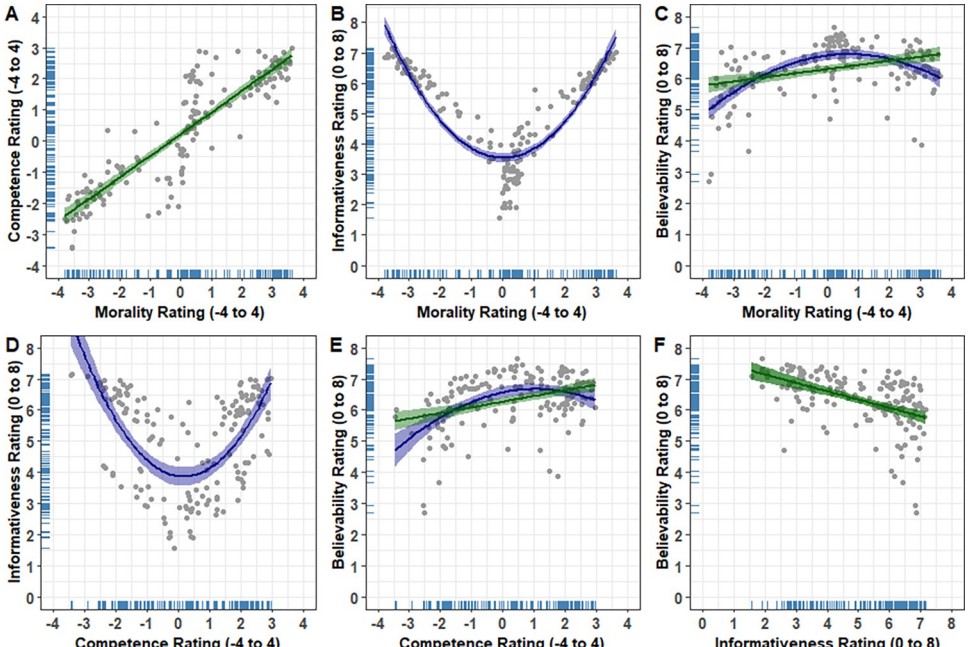

**Fig 3.** Scatter plots depicting the association between (A) morality and competence, (B) morality and informativeness, (C) morality and believability, (D) competence and informativeness, (E) competence and believability, and (F) informativeness and believability. Dot points represent the mean ratings for each behavior statement. The green lines show the linear trends and the blue lines show the quadratic trends. The shaded areas show the 95% confidence intervals. Rugs (i.e., the blue lines along the x- and y-axes) show distribution density.

**Table 2. Orthogonal polynomial regression (linear and quadratic) results for morality, competence, informativeness, and believability dimensions.**

|  | Outcome | Predictor | B | SE B | β | t | p |
|---|---|---|---|---|---|---|---|
| (A) | Competence | Morality | 19.51 | 0.81 | 0.89 | 23.98 | < .001 |
|  |  | Morality$^2$ | -0.92 | 0.81 | -0.04 | -1.13 | .262 |
|  |  | $F(2, 157) = 288.10, p < .001, R^2 = .79$ | | | | | |
| (B) | Informativeness | Morality | -1.09 | 0.67 | -0.06 | -1.63 | .105 |
|  |  | Morality$^2$ | 17.77 | 0.67 | 0.90 | 26.65 | < .001 |
|  |  | $F(2, 157) = 356.40, p < .001, R^2 = .82$ | | | | | |
| (C) | Believability | Morality | 3.72 | 0.70 | 0.34 | 5.33 | < .001 |
|  |  | Morality$^2$ | -5.20 | 0.70 | -0.48 | -7.46 | < .001 |
|  |  | $F(2, 157) = 42.06, p < .001, R^2 = .35$ | | | | | |
| (D) | Informativeness | Competence | -1.00 | 1.22 | -0.05 | -0.82 | .412 |
|  |  | Competence$^2$ | 12.31 | 1.22 | 0.63 | 10.07 | < .001 |
|  |  | $F(2, 157) = 51.05, p < .001, R^2 = .39$ | | | | | |
| (E) | Believability | Competence | 3.99 | 0.76 | 0.37 | 5.23 | < .001 |
|  |  | Competence$^2$ | -3.10 | 0.76 | -0.29 | -4.05 | < .001 |
|  |  | $F(2. 157) = 21.87, p < .001, R^2 = .22$ | | | | | |
| (F) | Believability | Informativeness | -5.29 | 0.75 | 1.53 | -7.05 | < .001 |
|  |  | Informativeness$^2$ | -0.96 | 0.75 | -2.03 | -1.28 | .204 |
|  |  | $F(2, 157) = 25.66, p < .001, R^2 = .25$ | | | | | |

*Note.* N = 160 behavior statements; B = unstandardized beta weights; SE B = standard errors of unstandardized beta weights; β = standardized beta weights.

behavior statements need to be believable to affect person impressions [5], the general believ-ability of the behavior statements should be advantageous to researchers using the corpus.

Researchers interested in the influence of specific dimensions on impression formation may need to control for the contribution of related dimensions. Our results indicate a range of linear and quadratic relationships between the morality, competence, informativeness, and believability dimensions. The morality and competence dimensions showed a positive linear relationship, indicating that more morally positive behavior statements were rated as more competent. This replicates prior research [29, 42, 43], and suggests a halo effect [10, 44] whereby favorable judg-ments on the morality dimension positively influence judgements on the competence dimension (or vice versa). The informativeness dimension showed a quadratic relationship to the morality and competence dimensions: behavior statements rated as more extreme in morality or compe-tence (i.e., extreme positive or extreme negative) were associated with an increase in informative-ness. These findings are consistent with an extremity bias, whereby more morally extreme information is given greater weight in impression formation [24, 45, 46].

The believability dimension showed positive linear relationships and quadratic relation-ships with the morality and competence dimensions. Behavior statements rated as more posi-tive in morality/competence were generally rated as more believable (than more negative statements), and more extreme (positive/negative) behaviors were associated with a decrease in believability. These relationships may be explained by people's expectations, in so far as peo-ple expect others to behave in positive and non-extreme ways [e.g., person positivity bias, see 47] so find such behaviors more believable. Our final test showed a strong negative linear rela-tionship between informativeness and believability, indicating that more informative behavior statements were also rated as less believable. Together, these findings make intuitive sense, sug-gesting that more unexpected and surprising behaviors, which are less believable, are consid-ered to be more informative [19, see also "frequency-weight" theories, 22]. The negative relationship between informativeness and believability is also consistent with recent research on misinformation (e.g., fake news and conspiracy theories). Even if low in believability, mis-information can be perceived to be 'informative if true', and therefore has the potential to strongly sway opinion [48, 49] and be widely shared online [50, see also 51].

To conclude, the present study provides a normed corpus of 160 contemporary behavior statements. The statements were rated by a large sample of judges ($N = 400$, with each behavior statement rated by 100 judges) on four dimensions relevant to impression formation: morality, competence, informativeness, and believability. Importantly, the different dimensions were non-independent; a range of linear and non-linear relationships between the dimensions were identified. Accounting for these relationships (e.g., statistically) can help researchers avoid drawing unwarranted conclusions. For example, researchers investigating the effect of compe-tence on impression formation may find their results are better explained by morality [e.g., see 52] or that the effect of a specific dimension is moderated by statement informativeness or believability. Given these considerations, we believe the corpus of behavior statements gener-ated in the present study will be valuable to researchers interested in impression formation.

## Acknowledgments

We are grateful to Martin Wood and David Kernot (Defence Science and Technology Group) for their valuable feedback on early drafts of the manuscript.

## Author Contributions

**Conceptualization:** Amy Mickelberg, Bradley Walker, Ullrich K. H. Ecker, Piers Howe, Andrew Perfors, Nicolas Fay.

**Data curation:** Amy Mickelberg.

**Formal analysis:** Amy Mickelberg.

**Funding acquisition:** Nicolas Fay.

**Investigation:** Amy Mickelberg, Bradley Walker, Ullrich K. H. Ecker, Nicolas Fay.

**Methodology:** Amy Mickelberg, Bradley Walker, Piers Howe, Andrew Perfors, Nicolas Fay.

**Project administration:** Amy Mickelberg, Bradley Walker, Nicolas Fay.

**Software:** Bradley Walker.

**Supervision:** Bradley Walker, Nicolas Fay.

**Visualization:** Amy Mickelberg, Nicolas Fay.

**Writing – original draft:** Amy Mickelberg, Bradley Walker, Ullrich K. H. Ecker.

**Writing – review & editing:** Amy Mickelberg, Bradley Walker, Ullrich K. H. Ecker, Piers Howe, Andrew Perfors.

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
