## [Decision Letter · Decision Letter 0]

5 May 2022

PONE-D-22-09306Impression Formation Stimuli: A Corpus of Behavior Statements Rated on Morality, Competence, Informativeness, and BelievabilityPLOS ONE

Dear Dr. Fay,

Thank you for submitting your manuscript to PLOS ONE. After careful consideration, we feel that it has merit but does not fully meet PLOS ONE’s publication criteria as it currently stands. Therefore, we invite you to submit a revised version of the manuscript that addresses the points raised during the review process. The two reviewers and I all share the same basic conclusion: the manuscript is of high quality but can use just a bit of cleaning up. Both reviewers provided some very nice and simple suggestions to make this manuscript just that much better. I ask that the authors please address the reviewer comments before this manuscript is accepted for publication (which I anticipate it will be once the revisions are completed). I do not anticipate sending this back out to review, but will rather make the final determination on my own. As none of the requests were particularly onerous, I anticipate a fast turn around both from the authors and from me. Well done!

We look forward to receiving your revised manuscript.

Kind regards,

Jeff Galak, PhD

Academic Editor

PLOS ONE

Journal Requirements:

"We are grateful to Martin Wood and David Kernot (Defence Science and Technology Group) for their valuable feedback on early drafts of the manuscript.  The research was supported by a PhD scholarship funded by the Defence Science and Technology Group of the Department of Defence (A.M.) and a Defence Science and Technology (DST) Group Collaborative Agreement 8382 (N.F., B.W., A.P. and P.H.)."

We note that you have provided funding information. However, funding information should not appear in the Acknowledgments section or other areas of your manuscript. We will only publish funding information present in the Funding Statement section of the online submission form. 

"The research was supported by a PhD scholarship funded by the Defence Science and Technology Group of the Department of Defence (A.M.) and a Defence Science and Technology (DST) Group Collaborative Agreement 8382 (N.F., B.W., A.P. and P.H.).  The funders had no role in the study design, data collection and analysis or decision to publish.  They did provide feedback on draft of the manuscript."

"A.M. is funded by a PhD scholarship funded by the Defence Science and Technology Group of the Department of Defence.  N.F., B.W., A.P. and P.H. were funded by the Defence Science and Technology (DST) Group under a Collaborative Agreement 8382. U.K.H.E. was supported by an Australian Research Council grant FT190100708."

We note that you have provided funding information. However, funding information should not appear in the Funding section or other areas of your manuscript. We will only publish funding information present in the Funding Statement section of the online submission form. 

"The research was supported by a PhD scholarship funded by the Defence Science and Technology Group of the Department of Defence (A.M.) and a Defence Science and Technology (DST) Group Collaborative Agreement 8382 (N.F., B.W., A.P. and P.H.).  The funders had no role in the study design, data collection and analysis or decision to publish.  They did provide feedback on draft of the manuscript."

Reviewers' comments:

Reviewer's Responses to Questions

**Comments to the Author**

1. Is the manuscript technically sound, and do the data support the conclusions?

Reviewer #1: Yes

Reviewer #2: Yes

2. Has the statistical analysis been performed appropriately and rigorously? 

Reviewer #1: Yes

Reviewer #2: Yes

3. Have the authors made all data underlying the findings in their manuscript fully available?

Reviewer #1: Yes

Reviewer #2: Yes

4. Is the manuscript presented in an intelligible fashion and written in standard English?

Reviewer #1: Yes

Reviewer #2: Yes

5. Review Comments to the Author

Reviewer #1: I enjoyed reading this manuscript, thank you for including me in the process of publishing it.

Since authors are native speakers, the text was clear and easy to follow (I am not a native English speaker).

As for the research itself, I have several, suggestions/comments.

First of, for whom did you construct the Corpus, only for English speaking countries? Would the statements hold a similar meaning when translated? If they are intended only for English speaking countries, do you expect the effect of different cultures (for e.g. UK and US)? Also, why was the research conducted in US and not Australia (since authors are from Australia)?

When the sample was recruited, were there any other socio-demographic questions (beside age and gender)? I believe that they might also affect the interpretation of the given statements, have you controlled gender and age in the analyses and if not, why not?

Why did you opt for three of the five morality dimensions, elaborate this in the text (line 113, page 12).

Line 115, page 12, authors state: "we generated five categories of statements" - who is we, authors? How did you generate these statements, please explain the process in more detail.

Why did you use 48x2 statements for morality and 20x2 for competence? Why not equal number of statements for both constructs?

Finally, you basically used a 9-point Likert scale which is rather difficult for respondents to estimate, is there a reason why?

I hope my suggestions are useful to you and I wish you good luck in your future endeavors.

Reviewer #2: This manuscript seems fairly straight-forward. I suppose my comments might be more like a 'wish list'.

For one, I'd have loved the literature review to do more to highlight how the four character facets used are similar to other concepts from the trustworthiness/people perception literature (e.g., warmth vs. competence; ability vs. integrity vs. benevolence, etc.). Many of us study similar things (at least from a measurement perspective) that use different terms.

The big thing I had a hard time with was that so many of the behaviors seem divorced from descriptions I'd ever want to use (i.e., if i was describing someone in a specific profession). I guess there must be cases where someone would want to describe peoples' non-professional behaviors (having forgotten to water the garden), but I never have. I suppose there must be someone else who'd use these kinds of statements. More broadly, I worry that some of these things will mean competence (or morality) or be more/less believable in certain contexts but I didn't see much of a discussion of context. So much emphasis in the trust literature is on 'trustworthiness in the context of X ... not just generally)

In the methods section, it might be nice to say something about why you chose 40 statements and 100 participants. Time and budget, I assume ... but also, SE? Also, was there a zero in the -4 to 4 questions? I assume so because you have 0 for the 0-8 options. I really wish you hadn't switched to the from -4 to +4 to 0 to 8; that's just confusing. I almost wonder if you shouldn't rescale for the table just so the numbers are consistent.

Why not order table 1 by ... something. Even if it's just the first column. I see there's an interactive table on OSF but ... still. Also, however you end up printing this, it'd be nice to repeat the column headers at the top of each page.

6. PLOS authors have the option to publish the peer review history of their article (what does this mean?). If published, this will include your full peer review and any attached files.

Reviewer #1: **Yes: **Tamara Jovanović

Reviewer #2: No

---

## [Author Response · Author response to Decision Letter 0]

17 May 2022

Responses to reviewers and editor documented in the cover letter.

---

## [Editor Report · Decision Letter 1]

20 May 2022

Impression Formation Stimuli: A Corpus of Behavior Statements Rated on Morality, Competence, Informativeness, and Believability

PONE-D-22-09306R1

Dear Dr. Fay,

We’re pleased to inform you that your manuscript has been judged scientifically suitable for publication and will be formally accepted for publication once it meets all outstanding technical requirements.

Kind regards,

Jeff Galak, PhD

Academic Editor

PLOS ONE
---

## [Editor Report · Acceptance letter]

27 May 2022

PONE-D-22-09306R1 

Impression Formation Stimuli: A Corpus of Behavior Statements Rated on Morality, Competence, Informativeness, and Believability 

Dear Dr. Fay:

I'm pleased to inform you that your manuscript has been deemed suitable for publication in PLOS ONE. Congratulations! Your manuscript is now with our production department. 

Kind regards, 

on behalf of

Dr. Jeff Galak 

Academic Editor

PLOS ONE